# A Complementary Metal-Oxide-Semiconductor Optoelectronic Analog Front-End Preamplifier with Cross-Coupled Active Loads for Short-Range LiDARs

**DOI:** 10.3390/s25041040

**Published:** 2025-02-10

**Authors:** Yunji Song, Yejin Choi, Dukyoo Jung, Seonhan Choi, Sung-Min Park

**Affiliations:** 1Division of Electronic & Semiconductor Engineering, Ewha Womans University, Seoul 03760, Republic of Korea; dbswl0828@ewhain.net (Y.S.); enun0515@ewha.ac.kr (Y.C.); seonhan.choi@ewha.ac.kr (S.C.); 2Graduate Program in Smart Factory, Ewha Womans University, Seoul 03760, Republic of Korea; 3College of Nursing, Ewha Womans University, Seoul 03760, Republic of Korea; dyjung@ewha.ac.kr

**Keywords:** active-load, APD, cross-coupled, LiDAR, preamplifier, sensors

## Abstract

In this paper, a CMOS optoelectronic analog front-end (AFE) preamplifier with cross-coupled active loads for short range LiDAR applications is presented, which consists of a spatially modulated P^+^/N-well on-chip avalanche photodiode (APD), the differential input stage with cross-coupled active loads, and an output buffer. Particularly, another on-chip dummy APD is inserted at the differential input node to improve the common-mode noise rejection ratio significantly better than conventional single-ended TIAs. Moreover, the cross-coupled active loads are exploited at the output nodes of the preamplifier not only to help generate symmetric output waveforms, but also to enable the limiting operations even without the following post-amplifiers. In addition, the inductive behavior of the cross-coupled active loads extends the bandwidth further. The proposed AFE preamplifier implemented in a 180-nm CMOS process demonstrate the measured results of 63.5 dB dynamic range (i.e., 1 µA_pp_~1.5 mA_pp_ input current recovery), 67.8 dBΩ transimpedance gain, 1.6 GHz bandwidth for the APD capacitance of 490 fF, 6.83 pA⁄√Hz noise current spectral density, 85 dB power supply rejection ratio, and 32.4 mW power dissipation from a single 1.8 V supply. The chip core occupies the area of 206 × 150 µm^2^.

## 1. Introduction

For the past decades, light detection and ranging (LiDAR) sensors have been frequently exploited in various applications such as advanced driver assistance systems, robots, remote sensing systems, drones, mobile phones, and elder-care systems for senile dementia patients [1,2,3]. Figure 1a shows the block diagram of a typical pulsed time of flight (ToF) LiDAR system that is well known for providing a number of advantages including low implementation cost and design simplicity when compared to other schemes such as amplitude/frequency-modulated-continuous-wave (AMCW or FMCW) methods. Although these other schemes estimate the target distances by the phase or frequency variations between the transmitted (i.e., START) pulses and the reflected (i.e., STOP) pulses, the ToF LiDAR sensors emit light signals from a laser-diode driver directly to a target (e.g., the senile dementia patient shown in Figure 1) and the reflected light pulses are received by the analog front-end (AFE) preamplifier in the receiver. The detection range to the target can be simply measured by the time-intervals between the START and STOP pulses. Among the blocks of the AFE circuits in a LiDAR receiver, the preamplifier becomes crucial in the overall performance determination of the LiDAR sensors in terms of gain, bandwidth, noise, power supply noise rejection (PSRR), dynamic range (DR), and power dissipation. In particular, the DR and PSRR characteristics should be wide and large enough for short-range LiDARs.

Previously, many research studies have been conducted for the realization of novel AFEs that are equipped with these characteristics. In particular, several differential AFEs have been suggested for better PSRR, such as pseudo differential configurations [4,5] or fully differential architectures [6,7,8]. However, the former consumes large chip areas and suffers from significant mismatches, while the latter mandates power-hungry post-amplifiers. Meanwhile, conventional optical receivers utilize bond-wires to integrate off-chip avalanche photodiodes (APDs) on PC-boards [9,10]. However, these bond-wires may not only increase the packaging cost significantly in the realization of multi-channel sensor arrays but also need electrostatic discharge protection diodes that usually deteriorate the bandwidth and noise characteristics.

In order to avoid the aforementioned issues effectively, in this work, on-chip APDs are realized in a standard CMOS technology. Also, a fully differential architecture is now proposed, thereby not only acquiring the symmetric output waveforms, but also helping to discard the requirements of power-hungry limiting amplifiers, as shown in Figure 1b. This configuration leads to the substantially reduced power dissipation in consequence.

This brief is organized as follows. Section 2 describes the circuit operations of the proposed AFE preamplifier with cross-coupled active loads. Section 3 demonstrates the measurement results. Then, a conclusion is provided in Section 4.

## 2. Circuit Description

### 2.1. On-Chip P^+^/NW APD

On-chip APDs were realized by utilizing a 180-nm CMOS process in this work. Figure 2 depicts the cross-section of the fabricated on-chip APD of which configuration shares the same structure as the P^+^/N-well (NW) APD reported in ref. [11].

The light enters through the optical window, generating electron-hole pairs at the P^+^/NW junction. The generated holes further create additional electron-hole pairs in the NW, leading to the avalanche effect. The contacts from the p-substrate and the NW are connected to ac-ground so that the effect of slow diffusion currents can be eliminated, thus leading to high operation speeds. The P^+^ contact is connected to the preamplifier, while a shielded dummy APD is linked to the other input node of the preamplifier for symmetry, as depicted in Figure 3.

The on-chip APDs take an octagonal shape to prevent premature edge breakdown [12], in which the diagonal length of the optical window is 40 µm and the resulting parasitic capacitance is 0.49 pF. Measured results demonstrate 2.72-A/W responsivity at 11.05 V reverse bias voltage with the avalanche breakdown of 11.1 V at a wavelength of 850 nm.

### 2.2. Proposed AFE Preamplifier

Figure 3a shows the schematic diagram of the AFE preamplifier with cross-coupled active loads that consist of PMOS loads (M3, M4) and NMOS source-followers (M5, R1 and M6, R2) with their gates cross-coupled, thereby enabling the voltage output swing to reach the supply voltage (VDD). As the light pulses enter the on-chip APD, the optical signals are converted to electrical currents ($i_{pd}$) and flow toward the gate of M1. The variable resistor ($R_{F}$) between the input and the output nodes serves as negative feedback, where a gain-control scheme is added with a 3-bit digital-to-analog converter (DAC), hence further extending the input dynamic range.

Figure 3b illustrates the schematic diagram of an f_T_-doubler, in which the input capacitance becomes the series connection of C_gs7_ and C_gs8_, thus resulting in an effectively halved capacitance, i.e., $\frac{1}{2}C_{gs7}$, under the assumption that M7 and M8 have the same size. Therefore, the cutoff frequency (f_T_) of this f_T_-doubler is given as follows:(1)$$f_{T}\equiv \frac{g_{m}}{2\pi C_{gs}}≅\frac{2g_{m7}}{2\pi C_{gs7}},$$
which certainly enables the bandwidth extension [13]. This f_T_-doubler is employed as an output buffer to boost the transimpedance gain further with no sacrifice of bandwidth performance. Also, R3 is the output resistance of 50 Ω for output impedance matching.

Figure 4 shows a simplified equivalent circuit of the AFE preamplifier, which is used to analyze the small signal in terms of the transimpedance gain.

Small signal analysis shows that the transimpedance gain at the negative output node (OUT-) is given as follows:(2)$$\begin{array}{c} \left(g_{m1}-\frac{1}{R_{F}}\right)v_{in}\;=\;-v_{o-}\left[\frac{1}{R_{F}}\;-\;g_{m3}\left(\frac{g_{m5}R_{1}}{1+g_{m5}R_{1}}\right)\right]\\ ∴v_{in}\;=\;-\frac{v_{o-}}{\left(g_{m1}\;-\;\frac{1}{R_{F}}\right)}[\frac{1}{R_{F}}\;-\;g_{m3}\left(\frac{g_{m5}R_{1}}{1\;+\;g_{m5}R_{1}}\right)]\\ ∴\frac{v_{o-}}{i_{pd}}\;=\;\frac{-R_{F}\left(g_{m1}\;-\;\frac{1}{R_{F}}\right)\left(1\;+\;g_{m5}R_{1}\right)}{g_{m1}\left(1\;+\;g_{m5}R_{1}\right)\;-\;g_{m3}g_{m5}R_{1}}≅\frac{-R_{F}g_{m1}\left(1\;+\;g_{m5}R_{1}\right)}{g_{m1}g_{m5}R_{1}}≅-R_{F}, \end{array}$$
provided that g_m1_ >> g_m3_ and that R_F_ is large enough to be smaller than 1/g_m1_.

Also, the drain current of M1 flows through M2 and then M4. Therefore, the current flowing through M2 is equal to gm1vin that is identical to -gm4vgs4. Then, the transimpedance gain at the positive output node (OUT+) is represented as follows:(3)$$\frac{v_{o+}}{i_{pd}}=\frac{-{\left(\frac{g_{m1}}{g_{m4}}\right)R}_{F}}{\left(g_{m1}-\frac{1}{R_{F}}\right)}\left[\frac{1}{R_{F}}-g_{m3}\left(\frac{g_{m5}R_{1}}{1+g_{m5}R_{1}}\right)\right]≅R_{F}\left(\frac{g_{m3}}{g_{m4}}\right)=R_{F},$$
provided that R_F_ is large and g_m5_R_1_ >> 1. The last term is acquired by assuming g_m3_ = g_m4_.

Meanwhile, the −3 dB bandwidth of the AFE preamplifier is simply estimated by using Miller’s theorem, which is given as follows:(4)$$f_{-3dB}≅\;\frac{1}{{2\pi R}_{in}C_{tot}}≅\frac{1}{{2\pi R}_{F}C_{tot}}\left(1+\frac{g_{m1}}{g_{m3}}\right),$$
where C_tot_ represents the total input capacitance at the gate of M1 including the photodiode capacitance (C_pd_) and the gate capacitance of M1. The input resistance (R_in_) is given as follows:(5)$$R_{in}=\frac{R_{F}}{\left(1+\left|A_{v1}\right|\right)}=\frac{g_{m3}R_{F}}{\left(g_{m1}+g_{m3}\right)}\;\;A_{v1}\equiv \;\frac{v_{o-}}{v_{in}}=\frac{-\left(g_{m1}-\frac{1}{R_{F}}\right)}{\left[\frac{1}{R_{F}}-g_{m3}\left(\frac{g_{m5}R_{1}}{1+g_{m5}R_{1}}\right)\right]}≅\frac{g_{m1}}{g_{m3}}$$
assuming that R_F_ is large and that g_m5_R_1_ >> 1.

Noise current spectral density of the AFE preamplifier is approximately given as follows:(6)$$\bar{I_{eq}^{2}}≅\frac{4kT}{R_{F}}+\left(\frac{4kT\Gamma g_{m1}+4kT\Gamma g_{m3}}{g_{m1}^{2}}\right)\left(\frac{1}{R_{F}^{2}}+\omega^{2}C_{tot}^{2}\right)+\left(\frac{4kT\Gamma g_{m5}+\frac{4kT}{R_{1}}}{g_{m5}^{2}R_{F}^{2}}\right)≅\frac{4kT}{R_{F}}+\frac{4kT\Gamma \left(g_{m1}+g_{m3}\right)\omega^{2}C_{tot}^{2}}{g_{m1}^{2}}.$$

It is clearly seen from Equation (6) that g_m1_ and g_m5_ should be maximized while g_m3_ should be lowered for noise minimization.

Figure 5 shows the layout of the proposed AFE preamplifier integrated with two on-chip P^+^/N-well APDs, where the chip core occupies the area of 206 × 150 µm^2^. DC simulations reveal that the AFE preamplifier consumes 12 mW from a single 1.8 V supply, excluding the output buffer (OB).

Figure 6 depicts the simulated frequency response of the AFE preamplifier, obtaining the transimpedance gain of 68 dBΩ and the −3 dB bandwidth of 1.86 GHz. However, a peak of 3 dB occurs near the bandwidth, which may be attributed to the inductive peaking characteristics of the feedback, i.e., R_F_. Also, the AFE preamplifier achieves the average noise current spectral density of 5.95 pA⁄√Hz within the bandwidth, which corresponds to the optical input sensitivity of −31.2 dBm for 10^−12^ bit-error-rate (BER) with the 2.72-A/W responsivity of the on-chip APD [12].

Figure 7 illustrates the simulated phase margin of the proposed AFE preamplifier which demonstrates the phase margin of 102° and the gain margin of 9.5 dB, thus ensuring the stability of the feedback system. It is noted that the voltage gain on the right *y*-axis represents the loop gain of the AFE preamplifier. The loop gain (Aβ) at low frequencies is less than unity because the open-loop voltage gain (A) is equal to ~g_m1_/g_m3_ (=4 in this work) while the value of the feedback factor (β) is small enough to lower the loop gain.

Also, the peaking at high frequencies is attributed not only to the complex conjugate poles in the open-loop gain, but also to a zero occurred from the cross-coupled active load. This may deteriorate the circuit stability, and therefore careful design is mandatory to avoid severe peaking larger than 3 dB in the frequency response.

Figure 8 shows the simulated eye diagrams of the proposed AFE preamplifier for the 100-µA_pp_ input current at various data rates of 100 Mb/s, 500 Mb/s, and 1 Gb/s. Here, it is clearly seen that the output amplitudes of the eye diagrams are almost identical.

Figure 9 depicts the simulated pulse response of the AFE preamplifier for the different input currents (from 1 µA_pp_ to 1.2 mA_pp_), which confirms that the wide input dynamic range of 61.5 dB is acquired with the aid of a 3-bit DAC gain-control scheme. In particular, the cross-coupled active loads consisting of PMOS transistors with NMOS source-followers contribute to the wide input DR characteristics due to their limiting operations.

Meanwhile, an f_T_-doubler is exploited as the OB for the FD-TIA, hence helping not only to achieve the DC offset cancellation at the output nodes, but also to obtain the symmetric output signaling [13]. Therefore, almost identical amplitudes can be obtained at the differential output pulses.

## 3. Chip Implementation and Measurements

The proposed AFE preamplifier chips were implemented using a 180 nm CMOS process. Figure 10 shows the chip photo of the proposed AFE preamplifier and its test setup including the optical testing instruments. The chip core occupies an area of 206 × 150 µm^2^ and dissipates the DC power of 32.4 mW from a single 1.8 V supply.

Figure 11 demonstrates the measured frequency response of the AFE preamplifier together with the simulation results, in which the transimpedance gain of 67.8 dBΩ and the bandwidth of 1.6 GHz were measured using a network analyzer (E5071C, Keysight, Santa Rosa, CA, USA) with an input power level of −60 dBm. Here, a zero occurs at a rather low frequency, hence leading to a peak at 1 GHz and extending the bandwidth in consequence.

Figure 12 shows the measured output noise voltage of the AFE preamplifier, where the inherent noise voltage of the instrument (Agilent DCA 86100D, Keysight, Santa Rosa, CA, USA) was 0.677 mV_RMS_. Then, the equivalent noise current spectral density is estimated to be 6.83 pA⁄√Hz, which corresponds to the optical sensitivity of −30.9 dBm for 10^−12^ BER with the 2.72-A/W responsivity of the on-chip APD. The discrepancy from the simulations might be attributed to the equivalent circuit model of the on-chip APD.

Figure 13 compares the measured PSRR characteristics of the AFE preamplifier with the simulation results, demonstrating 85 dB PSRR within the bandwidth. It is clearly seen that the FD-TIA shows far better PSRR than the single-ended TIA configuration. The discrepancy of 5 dB higher from the simulations may be attributed to the chip layout, in which the power and ground lines were separated for better isolation.

Finally, the optical pulse responses of the AFE preamplifier were measured with a laser driver (Seed LDD; Notice Ltd., Anyang, Republic of Korea) utilizing a laser diode (Qphotonics, Ann Arbor, MI, USA) at a wavelength of 850 nm. Here, the generated optical pulses show the pulse width of 10 ns. Then, the output waveforms were measured with an oscilloscope (DSOX1202A, Keysight, Santa Rosa, CA, USA).

Figure 14 shows the optically measured pulses of the AFE preamplifier for input currents of 15 µA_pp_ and 1.5 mA_pp_, representing the minimum and maximum values. Due to equipment limitations, the minimum measurable input current was 15 µA_pp_. The peak amplitudes of each single-ended output voltage were measured to be 40 mV_pp_ and 1.56 V_pp_ with a 50 Ω termination at the other output node.

Table 1 compares the performance of the proposed AFE preamplifier with the prior arts. Ref. [14] introduced a TIA design based on a cascaded structure combining a current buffer and a voltage buffer. However, this configuration leads to increased power consumption and restricted gain performance.

Ref. [15] demonstrated a common-gate input current-mirror amplifier (CGCMA) to enlarge the output voltage swings and improve the accuracy. However, the CGCMA re-quired not only an external FPGA for programmable gain control, but also an on-chip high-pass filter that caused a large chip area. In addition, its narrow bandwidth could not avoid the long-tail of the output pulses.

Ref. [16] presented a 16-channel transimpedance amplifier (TIA) array using 16 off-chip InGaAs p-i-n photodiodes operating at a 1550-nm wavelength, where a volt-age-mode CMOS feedforward TIA was realized to acquire high transimpedance gain and low noise characteristics. However, the off-chip photodiode array led to the hardware complexity in multi-channel configurations, and its single-ended architecture resulted in poor CMRR characteristics.

Ref. [17] proposed a power-on-calibration technique not only to enhance the accuracy of transimpedance gain, but also to eliminate the mismatches of multi-channel TIAs. In particular, a noise canceling scheme was exploited to improve the circuit sensitivity. However, the resistor arrays should be precisely controlled and an on-chip high-pass filter was also mandatory to reduce the noise. In addition, the former required a very careful layout, and the latter resulted in a large chip area. Furthermore, optically measured results were not demonstrated.

Ref. [18] suggested an approach of transimpedance-to-noise optimization to achieve the minimum noise characteristics. Nonetheless, the input dynamic range characteristic was significantly narrow (i.e., 25 dB) for LiDAR sensor applications. It might be attributed to its high transimpedance gain and the absence of gain control.

Ref. [19] presented a reconfigurable TIA, in which a three-stage core amplifier helped to boost the open-loop gain and its stability. It also demonstrated the maximum detectable current of 5 mA_pp_ with the aid of gain control scheme. However, the following post-amplifier consumed notably high power.

In this work, superior performance is provided with the efficient single-to-differential conversion from the input stage without the requirement of area-consuming passive low-pass filters, thus leading to a small chip area, high PSRR, low noise, and wide DR characteristics. It also demonstrates the capability of discarding power-hungry limiting amplifiers, and hence improving the power efficiency in the system levels.

The high power supply rejection ratio (PSRR) and wide dynamic range (DR) characteristics of this work are essential for achieving robust performance in environments with varying noise levels, i.e., common scenarios in applications such as autonomous vehicles and industrial automation. Also, low power consumption and a compact chip area align with the growing need for energy-efficient and cost-effective solutions in multi-channel LiDAR sensor systems. In addition, the realization of on-chip CMOS avalanche photodiodes (APDs) not only simplifies the sensor design by eliminating the need of external photodiodes but also reduces manufacturing and packaging costs. This is especially relevant for emerging LiDAR applications in consumer electronics, such as smartphones and drones, where minimizing size and cost are critical. Furthermore, as the markets increasingly demand the LiDAR solutions to be capable of supporting higher resolution and faster data acquisition, the wide bandwidth and low noise characteristics of the proposed AFE preamplifier ensure reliable operations at high data rates. These features are crucial to acquire accurate depth sensing and mapping in next-generation LiDAR systems used for augmented reality, robotics, and advanced driver assistance systems.

## 4. Conclusions

We have presented an AFE preamplifier realized in a 180 nm CMOS technology for the applications of short-range LiDAR sensors. It provides a fully differential architecture to achieve high PSRR characteristics and also exploits cross-coupled active loads to enlarge the dynamic range significantly along with the low power and small area characteristics. The measured results of the test chips demonstrate 67.8 dBΩ transimpedance gain, 1.6 GHz bandwidth, −27.8 dBm sensitivity for 10^−12^ BER, 63.5 dB dynamic range, 85 dB PSRR, and 32.4 mW power consumption from a single 1.8 V supply. Conclusively, it is certain that this work shows a potential for low-cost CMOS solutions in the applications of short-range LiDAR sensors.

## Figures and Tables

**Figure 1 sensors-25-01040-f001:**
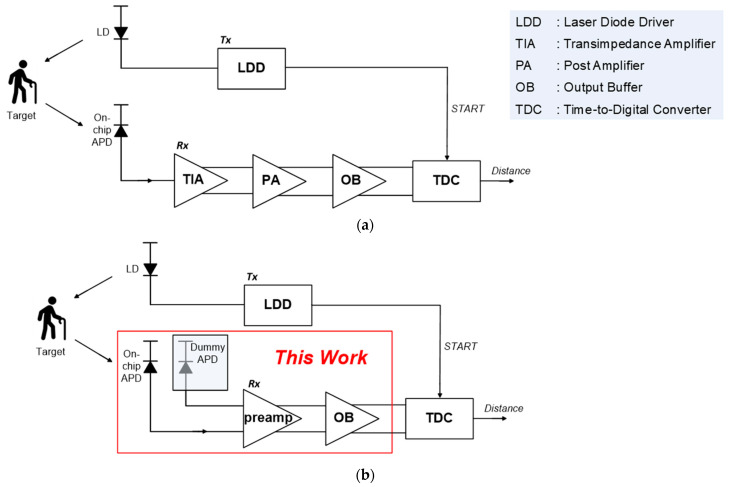
Block diagrams of (**a**) a typical short-range LiDAR system and (**b**) the proposed AFE preamplifiers for short-range LiDAR system.

**Figure 2 sensors-25-01040-f002:**
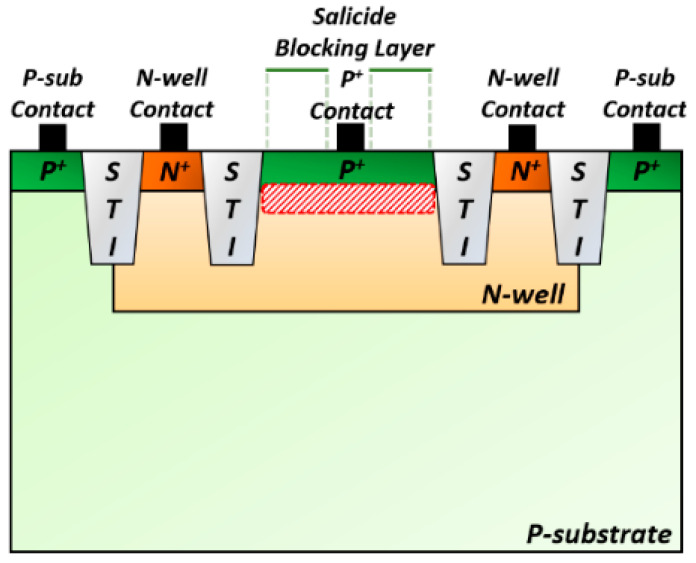
Cross-section of the on-chip P^+^/N-well APD.

**Figure 3 sensors-25-01040-f003:**
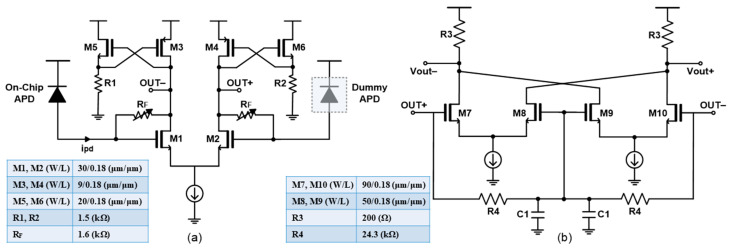
Schematic diagrams of (**a**) the proposed preamplifier and (**b**) the f_T_-doubler as an output buffer.

**Figure 4 sensors-25-01040-f004:**
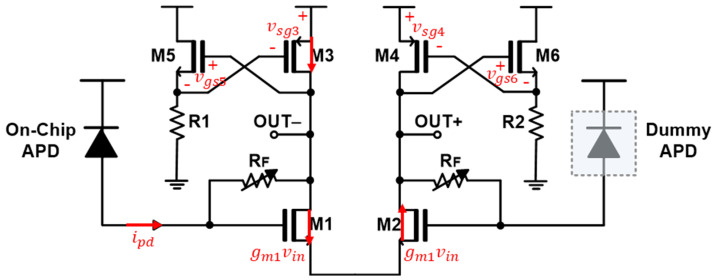
Simplified schematic diagram of the AFE preamplifier.

**Figure 5 sensors-25-01040-f005:**
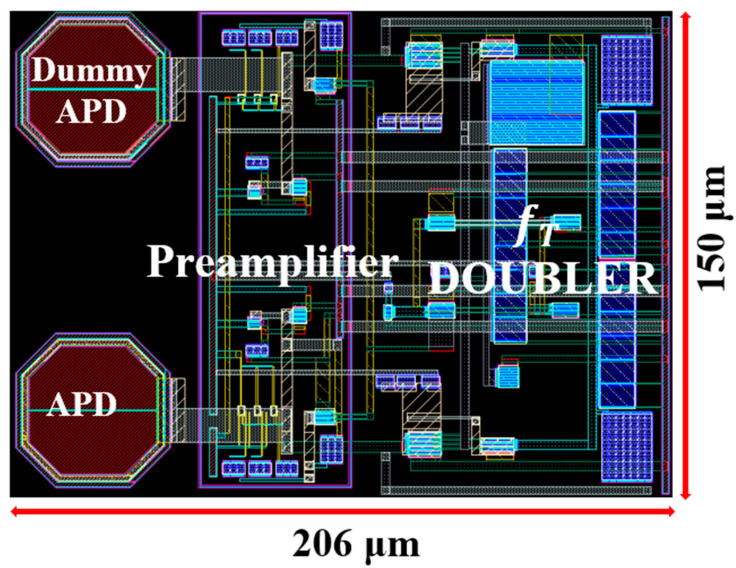
Layout of the proposed AFE preamplifier.

**Figure 6 sensors-25-01040-f006:**
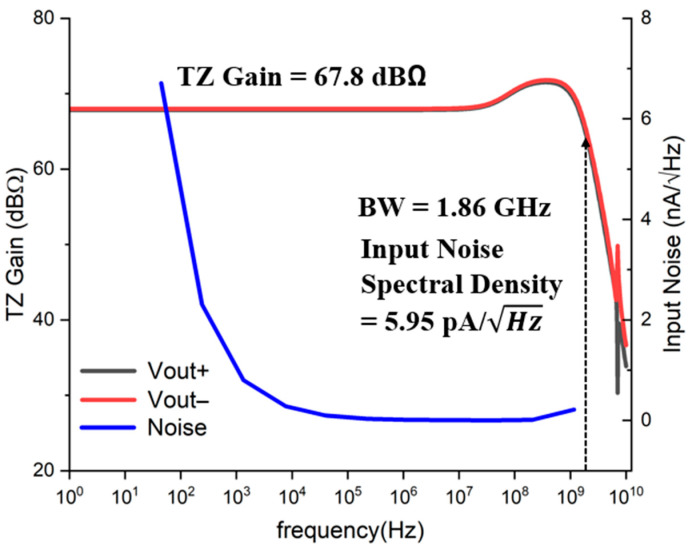
Simulated frequency response of the AFE preamplifier.

**Figure 7 sensors-25-01040-f007:**
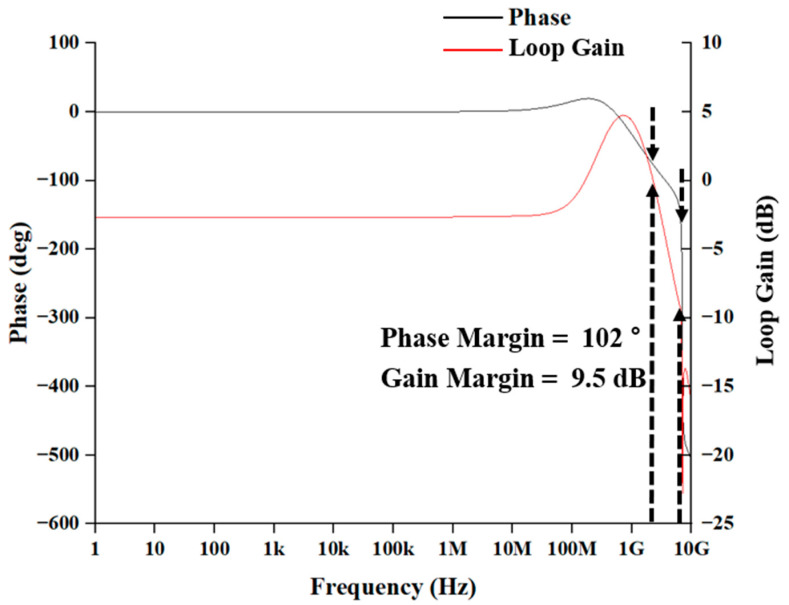
Simulated phase margin of the AFE preamplifier.

**Figure 8 sensors-25-01040-f008:**
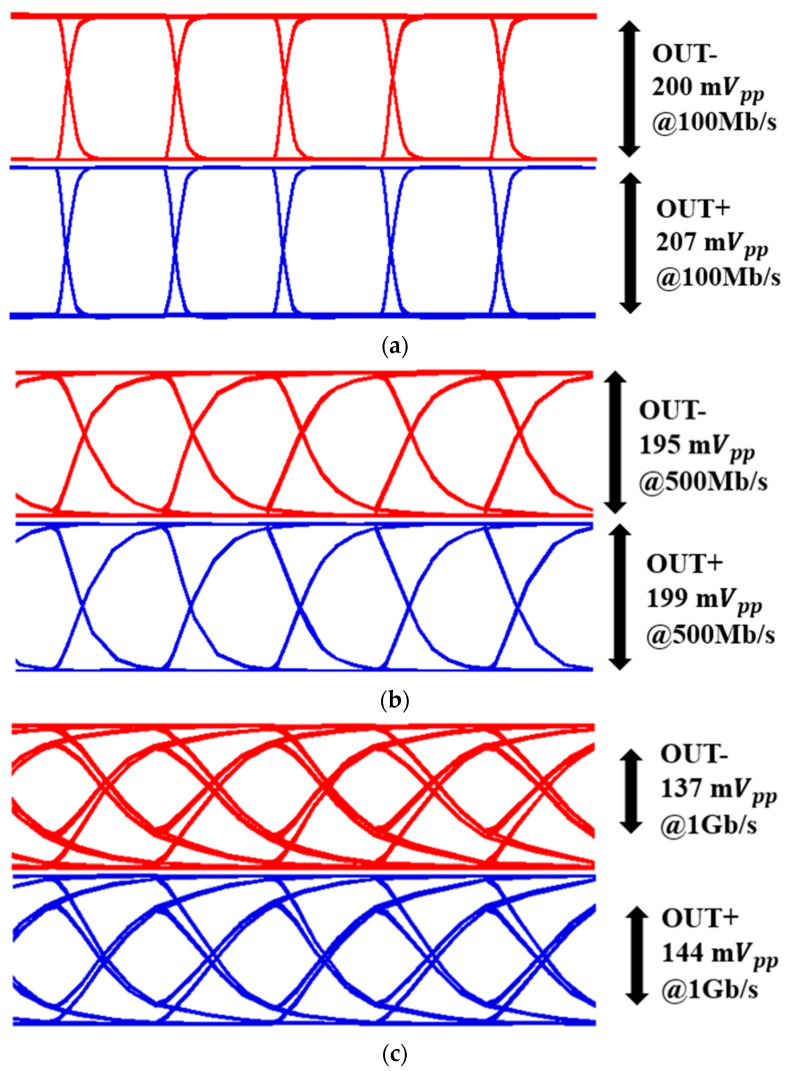
Simulated eye diagrams of the AFE preamplifier for 100 μ$A_{pp}$ input currents at different data rates of (**a**) 100 Mb/s, (**b**) 500 Mb/s, and (**c**) 1 Gb/s.

**Figure 9 sensors-25-01040-f009:**
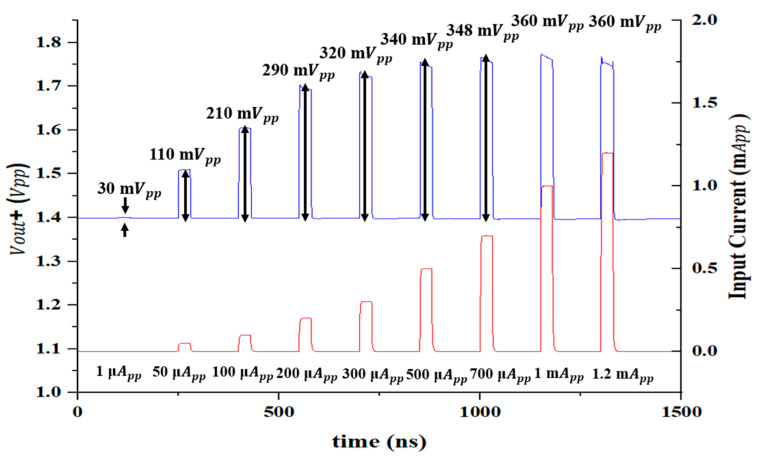
Simulated output pulses of the AFE preamplifier for different input currents.

**Figure 10 sensors-25-01040-f010:**
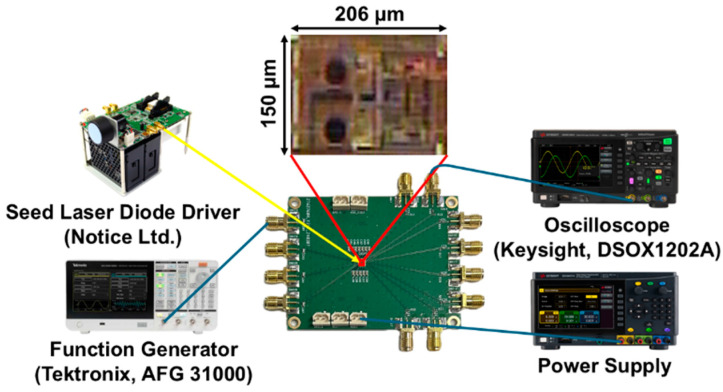
Chip photo of the proposed AFE preamplifier and its test setup (inc. optical test).

**Figure 11 sensors-25-01040-f011:**
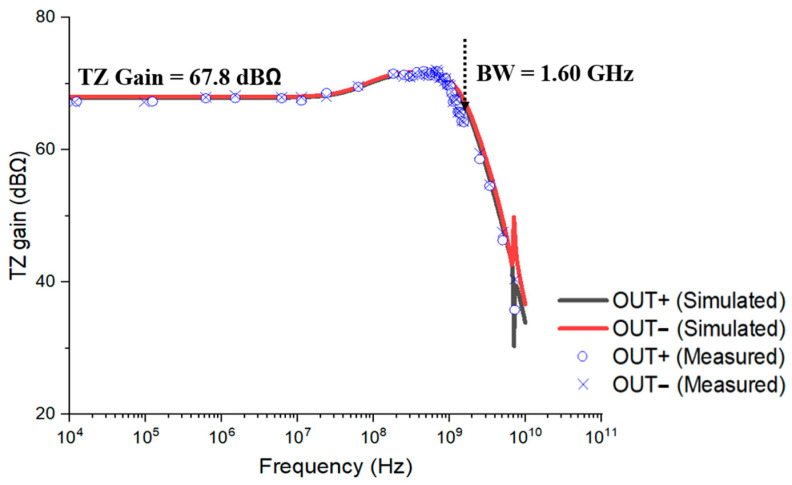
Measured frequency response of the AFE preamplifier.

**Figure 12 sensors-25-01040-f012:**
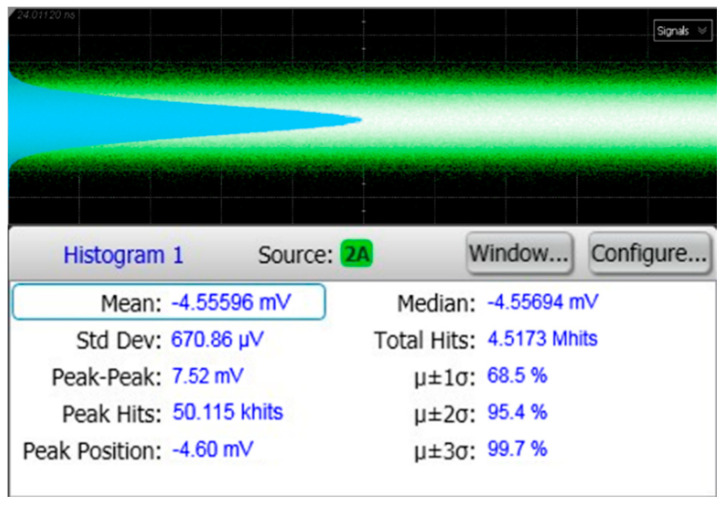
Measured output noise voltage of the AFE preamplifier.

**Figure 13 sensors-25-01040-f013:**
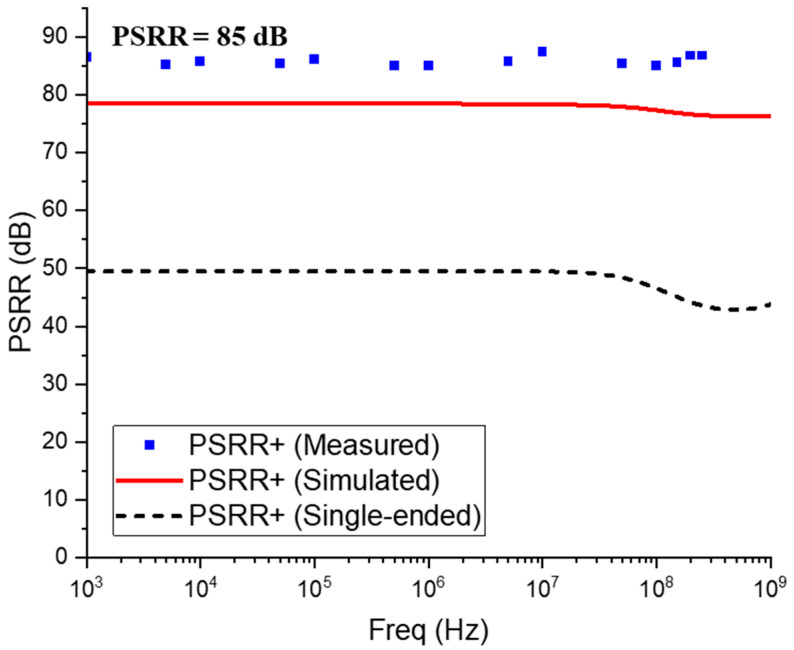
Simulated vs. measured PSRR of the AFE preamplifier.

**Figure 14 sensors-25-01040-f014:**
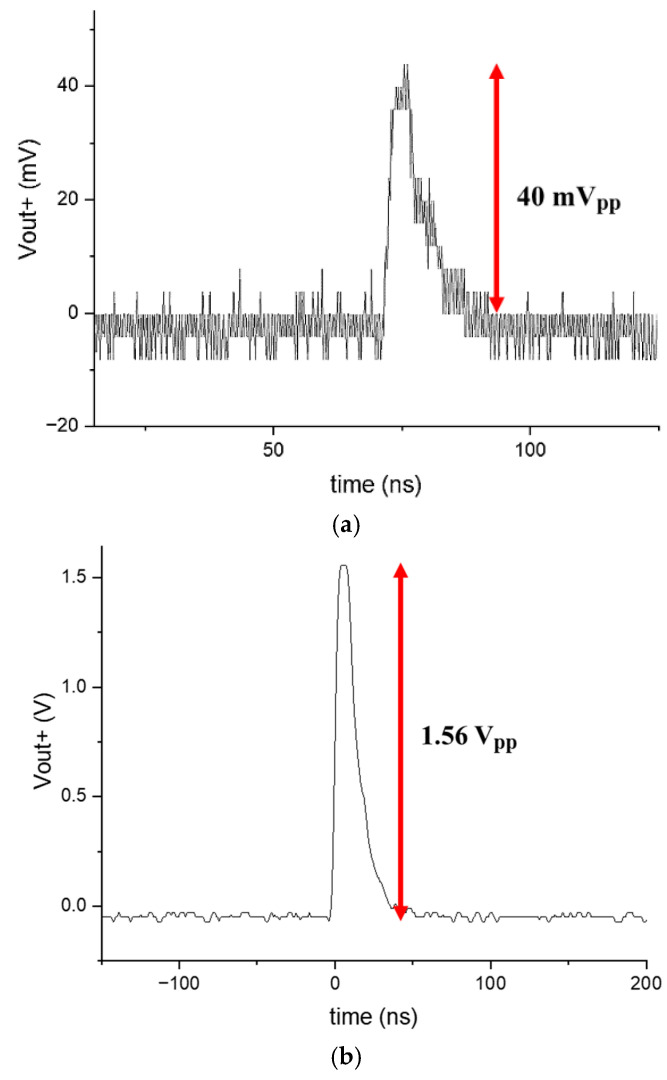
Optically measured pulses of the AFE preamplifier for (**a**) 15 μA_pp_ and (**b**) 1.5 mA_pp_ input currents (pulse width: 10 ns).

**Table 1 sensors-25-01040-t001:** Performance comparison with the recently published AFE preamplifiers for LiDAR sensors.

Parameters		[14]	[15]	[16]	[17]	[18]	[19]	This Work
CMOS technology (nm)		65	180	180	180	110	180	180
PD	Type	Off-chip (Equiv. PD)	Off-chip (APD)	Off-chip (PIN)	Off-chip (Equiv. PD)	Off-chip (Equiv. PD)	Off-chip (Equiv. PD)	On-chip (APD)
	C_pd_ (pF)	0.5	1.2	1.2	1.5	1.0	1.5	0.5
	Responsivity (A/W)	N/A	5	50@200 V	N/A	0.9@0.5 V	N/A	2.72
	Wavelength (nm)	N/A	905	905	N/A	1550	N/A	850
Input configuration		VCII	CGCMA ^‡^	VCF ^‡^	INV ^‡^	Diff.	Diff. (S2D) ^ξ^	Diff.
Max. TZ gain (dBΩ)		52	100	76.3	86	99	100	67.8
Gain control		No	Yes	Yes	Yes	No	Yes	Yes
Bandwidth (MHz)		1100	110	720	240	340	260	1600
Min. detectable current (mA_pp_)		N/A	1.0 @ SNR = 5	1.14 @ SNR = 6.3	0.5 @ SNR = 10	1.7 @ SNR = 10	0.54 @ SNR = N/A	1 @ SNR = 7
Max. detectable current (mA_pp_)		N/A	10 ^†^	1.1	1	0.03	5	1.5
Dynamic range (dB)		42	66	60	66	25	79.3	63.5
Noise current spectral density (pA/√Hz)		22	2.21	6.3	3.1	3.67	3.3	6.83
PSRR (dB)		N/A	N/A	N/A	N/A	87	61 *	85
Power dissipation per channel (mW)		55.3 @1.2 V	21 (8 @ TIA)	29.8	39.6 @ 3.3 V	41	153 @3.3 V	32.4
Chip area (mm^2^)		0.142	0.356	5.5 (16 ch.)	0.165	0.023	0.012 (TIA only)	0.031

^‡^ single-ended, ^†^ with FPGA-based gain control, ^ξ^ with a low-pass filter (single-to-differential or S2D) and a post-amplifier required, * simulated, N/A (not available)

## Data Availability

Data are contained within the article.

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
