# Peer review of "A Complementary Metal-Oxide-Semiconductor Optoelectronic Analog Front-End Preamplifier with Cross-Coupled Active Loads for Short-Range LiDARs"

_sensors, 2025, doi:10.3390/s25041040_

Round 1
Reviewer 1 Report
Comments and Suggestions for Authors
Please, see the attached file

Author Response
Dearest Reviewer 1,
Please find the attached answer sheet.
Best regards,
Sung Min Park

Reviewer 2 Report
Comments and Suggestions for Authors
This manuscript reported a CMOS analog-frontend preamplifier for short range LiDAR. It adopts cross-coupled active load structure, which not only helps to generate symmetrical output waveform and realize amplitude limiting operation, but also can further expand bandwidth. At the same time, on-chip dummy APD is inserted into the differential input node to significantly improve the common-mode noise rejection ratio. Through simulation and measurement, the preamplifier shows excellent performance. Overall, this work has some new and interesting results, and I suggest its acceptance after addressing the following concerns.
1. Figure 1(b) is given but not described in the article, can you describe it? Can the abbreviations in Figure 1 be marked with their full names?
2. The way in which the input dynamic range is extended is proposing a gain control scheme with a 3-bit digital-to-analog converter. There is not enough explanation of the implementation mechanism of the extended input dynamic range here. How does the 3-bit digital-to-analog converter be added to the preamplifier and how does the input dynamic range be extended?
3. Your article gives Figure 3(b), but there is no language description, can you describe the implementation mechanism of Figure 3(b) as an output buffer? Why do three resistors appear in parallel, can they be directly represented by a resistor?
4. The first letter of the title in Figure 4 need not be bold.
5. Line 144 is "~3 dB", right?
6. As for the principle of bandwidth widening by cross-coupled active load inductance, only the phenomenon is mentioned, and no in-depth explanation is given. It is suggested to add detailed mathematical derivation and physical analysis, combined with circuit model, to explain the interaction between inductance characteristics and other circuit parameters, such as analyzing the influence of inductance on signal transmission at different frequencies through small signal model, so as to provide more solid theoretical support for circuit design.
7. What is the point of the same output amplitude of the eye map? Can you describe it briefly?
8. Line 177 is "~1 GHz", right?
9. Does the single-ended TIA in Figure 12 have a specific model?
10. In addition to the technical performance advantages, can you analyze the advantages of the preamplifier in practical application scenarios combined with the market demand and development trend of short-range LiDAR?
Author Response
Dearest Reviewer 2,
Please find the attached answer sheet.
Best regards,
Sung Min Park

Round 2
Reviewer 1 Report
Comments and Suggestions for Authors
The manuscript has been improved, but there are still relevant issues/errors that must be properly addressed/corrected.
1. First, concerning the calculations of the closed loop small signal transimpedance gain of the TIA, the response of the authors to the observations done in the first round of the revision is not satisfactory. The authors have eliminated the details of the evaluation of the vo+ voltage, but basically they didn’t modify, for instance, eq. 2 which, in my view, contains an error that has been reported in the previous revision round. The authors, in their response, state that eq. (1)-(6) might be misleading, but they didn’t modify (except for the apparent dimensional error, that has been corrected) or clarify these equations and, especially, they didn’t comment about the reported error in the term 2+gm5*R1 in eq. (1).
2. Concerning the phase margin evaluation, it is not clear what has been reported in Fig. 7. What exactly is the voltage gain that is plotted against frequency in this figure ? It should be the loop gain of the circuit, in order to evaluate the phase margin, but how is it possible that the loop gain in the bandwidth of the circuit is less than 1 (or 0dB) ? In this case, the closed loop gain cannot be equal to RF (i.e. the inverse of the feedback factor) and the feedback is absolutely ineffective. Moreover, the voltage gain plotted in Fig. 7 exhibits a peak: so, the open loop gain exhibits already complex conjugate poles, before the feedback loop is closed, which is definitely unusual. Last, a phase margin of 102° cannot correspond to a circuit which exhibits a peak in the closed loop gain.
3. The part concerning the evaluation of the closed loop bandwidth of the TIA (eq. 4) is not convincing. The authors claim that the “Since the architecture of the AFE preamplifier is a voltage-mode shunt-feedback TIA, the bandwidth can be simply analyzed by using the open-circuit time-constant method”: this is arguable, since in a feedback amplifier, the open loop poles (which can be evaluated with the open-circuit time-constant method) the poles are moved away from the open-loop position and can become complex conjugate, giving rise, possibly to the peaking in the closed loop response of the amplifier, as happens in this case. Thus, in this case, it is not possible to express the closed loop bandwidth simply in terms of the inverse of a time constant, because the closed loop poles are complex conjugate, as clearly shown by the peaking in the closed-loop response.
4. In their response the authors claim that “Fig. 10 shows the frequency response of the proposed AFE preamplifier, where the frequency of the input current varies as a sinewave. In this AC simulation, the amplitudes of input currents are irrelevant. Only the DC offset current might be included for practical use.” Indeed, the observation made in the previous revision round was not about the simulations, but about the measured points that have been reported on the simulations in Fig. 10 (now Fig. 11). For these points, it is relevant to understand what are the measurement conditions that have been used, which was the purpose of the observation raised in the previous revision round, reported in the following: “What are the conditions in which Fig. 10 has been obtained ? What is the amplitude of the input current ? How this current has been injected into the TIA ?” All these questions are referred to the measured points, not to the simulations.
Comments on the Quality of English LanguageNo comments.
Author Response
Dearest Reviewer 2,
Please find the attached file that is the answer sheet for your valuable comments.
Best regards,
Sung Min Park

Reviewer 2 Report
Comments and Suggestions for Authors
The paper is acceptable without further revision.
Author Response
Dearest Reviewer,
Thanks a lot for your help.
Best regards,
Sung Min Park